# Immersive Virtual Reality during Robot-Assisted Gait Training: Validation of a New Device in Stroke Rehabilitation

**DOI:** 10.3390/medicina58121805

**Published:** 2022-12-07

**Authors:** Charles Morizio, Maxence Compagnat, Arnaud Boujut, Ouiddad Labbani-Igbida, Maxime Billot, Anaick Perrochon

**Affiliations:** 1HAVAE Laboratory, UR20217, University of Limoges, F-87000 Limoges, France; 2Department of Physical Medicine and Rehabilitation, University Hospital Center of Limoges, F-87000 Limoges, France; 33iL Groupe, F-87015 Limoges, France; 4XLim Institute, UMR CNRS 7252, University of Limoges, F-87068 Limoges, France; 5PRISMATICS Lab (Predictive Research in Spine/Neuromodulation Management and Thoracic Innovation/Cardiac Surgery), Poitiers University Hospital, F-86000 Poitiers, France

**Keywords:** virtual reality, head-mounted display, robot-assisted gait training, gait rehabilitation, cybersickness

## Abstract

*Background and objective:* Duration of rehabilitation and active participation are crucial for gait rehabilitation in the early stage after stroke onset. Virtual reality (VR) is an innovative tool providing engaging and playful environments that could promote intrinsic motivation and higher active participation for non-ambulatory stroke patients when combined with robot-assisted gait training (RAGT). We have developed a new, fully immersive VR application for RAGT, which can be used with a head-mounted display and wearable sensors providing real-time gait motion in the virtual environment. The aim of this study was to validate the use of this new device and assess the onset of cybersickness in healthy participants before testing the device in stroke patients. *Materials and Methods:* Thirty-seven healthy participants were included and performed two sessions of RAGT using a fully immersive VR device. They physically walked with the Gait Trainer for 20 min in a virtual forest environment. The occurrence of cybersickness, sense of presence, and usability of the device were assessed with three questionnaires: the Simulator Sickness Questionnaire (SSQ), the Presence Questionnaire (PQ), and the System Usability Scale (SUS). *Results:* All of the participants completed both sessions. Most of the participants (78.4%) had no significant adverse effects (SSQ < 5). The sense of presence in the virtual environment was particularly high (106.42 ± 9.46). Participants reported good usability of the device (86.08 ± 7.54). *Conclusions:* This study demonstrated the usability of our fully immersive VR device for gait rehabilitation and did not lead to cybersickness. Future studies should evaluate the same parameters and the effectiveness of this device with non-ambulatory stroke patients.

## 1. Introduction

Stroke is a leading cause of disability considering that 63% of patients have no independent walking function on admission to rehabilitation and this rate is 36% at discharge of hospitalization [1]. About 80% of stroke patients experience altered walking abilities 3 months after onset [2]. Walking disorders can result from deficits in muscle strength, motor control, and balance [2,3]. As a result, early recovery of walking abilities is a major concern for stroke patients insofar as it is a determining activity for autonomy, social participation, and quality of life [4]. Gait rehabilitation for stroke patients needs to start as early as possible to promote the recovery of walking ability [5]. Among the different approaches to gait rehabilitation, robot-assisted gait training (RAGT) is specifically offered to non-ambulatory patients, providing a complex gait cycle with body weight support [6].

RAGT appears to provide task-specific rehabilitation without delay and with more intensity than conventional methods [7] and may therefore promote brain plasticity [8]. The use of RAGT has been shown to improve the effectiveness of gait rehabilitation after stroke [6,7].

However, a lack of motivation and entertainment can lead to decreased rehabilitation time and RAGT effectiveness [9]. To improve the playfulness and adherence of patients during RAGT, several studies have investigated the use of non-immersive virtual reality (VR) such as flat screens [9,10,11]. They showed high acceptability and motivation [9], improved dual task performance [10], and increased balance and gait capabilities [11]. The use of non-immersive VR with RAGT has led to motor performance improvement associated with neural plasticity [11]. In addition, a higher level of immersion could increase the intrinsic motivation to participate, reduce the perceived time spent in rehabilitation, and refocus the patient on the task by reducing distractors [12,13].

The level of immersion varies according to VR definition, VR equipment, and virtual content [14]. VR is defined as “the use of interactive simulations created with computer hardware and software to present users with opportunities to engage in environments that appear and feel similar to real-word objects and events” [15]. This definition does not distinguish between non-immersive systems (3D environment displays on 2D monitors), semi-immersive, and fully immersive VR (e.g., head-mounted display (HMD), headset alone), which provide more highly engaging and realistic environments [14,16]. The use of controllers allows interaction in the virtual environment and improves the immersion by a sense of presence [13], which is also a psychological state in which the individuals feel that they belong to the virtual environment [17]. Sense of presence is modulated by several technical features of the VR tool such as the tracking level or field of view [13]. The degree of interaction with the environment, the fidelity and realism of the movements, and the landscape will influence the sense of presence. Embodiment within a virtual avatar can also encourage the feeling that the environment is ‘real’. It bears mentioning that embodiment in an avatar is modulated by the quality of synchronization between the avatar and the user’s actions [18]. Fully immersive VR via HMD is an innovative and playful approach that enables rehabilitation to focus on specific tasks, such as walking in controlled environments.

Few studies have investigated the use of fully immersive VR during walking tasks in neurological disease [12,19]. Winter et al. [12] designed a virtual scenario as a game to promote adherence and motivation of stroke patients during treadmill sessions with a playful scenario for ambulatory patients. Kim et al. [19] used a headset alone to propose a virtual city scene to older adults and individuals with Parkinson’s disease during treadmill training. They found good acceptability without alteration of balance abilities after each VR session. In addition, the level of stress was lower after exposure to the virtual environment. These studies focusing on ambulatory patients support the use of fully immersive VR in gait rehabilitation [12,19].

However, some users may develop symptoms such as motion sickness, also known as “cybersickness”. At present, the mechanisms involved in cybersickness are not yet fully understood and different theories have been proposed [20,21]. Cybersickness may be due to a sensory conflict between the signals that provide information on orientation and body movement. Since users are not actually moving, their proprioceptive and vestibular organs fail to provide movement cues and lead to sensory mismatches [22]. The virtual environments with the lowest occurrence of side effects are those using physical displacement (linked to the individual’s movements) in a scene with little visual stimulation [23].

Today, there is no fully immersive VR rehabilitation device combined with RAGT suitable for non-ambulatory stroke patients and which could promote patient adherence and active participation due to the presence of biofeedback and its playful aspect [12]. It appears very interesting to develop a new device that offers high-quality immersion and gait simulation to promote patient adherence while reducing the risk of side effects. The aim of this work was to create a new gait rehabilitation device using RAGT and fully immersive VR and to assess the sense of presence, side effects, and the usability in healthy participants.

## 2. Materials and Methods

Thirty-eight healthy people (20 women) among physiotherapy students and healthcare professionals were recruited for this study. The inclusion criteria were as follows: adults (>18 years old) and having completed a Motion Sickness Susceptibility Questionnaire Short Form (MSSQ-Short) ≤ 26 in order not to involve individuals susceptible to developing motion sickness [24,25]. The exclusion criteria were: (i) pathologies that do not allow use of VR (cerebellar, vestibular, and epileptic disorders) and (ii) significant visual or visual–perceptual deficits. All of the subjects gave their informed consent to participate in the study. The study was conducted in accordance with the Declaration of Helsinki, and the protocol was approved by the ethics committee of the University Hospital Center of Poitiers (N° F20220719114034) on 19 July 2022.

### 2.1. Robot-Assisted Gait Training and VR-Device

We used the RAGT device Gait Trainer (GT) GT1 (Reha-Stim^®^ Medtec Inc. New York, NY, USA), which is a complex gear system providing a gait-like movement with two footplates and partial or total body weight support [26]. It simulates the stance and swing phase with a ratio of 60% to 40% between the two phases and allows velocity adjustment (from 0.14 to 1.1 m/s). It provides a controlled propulsion system that adjusts its output according to the patient’s effort and manages the control of the center of mass. [26].

We developed an application with Unity software (Unity Technologies^®^, San Francisco, CA, USA) on a Microsoft Windows 10-based computer system that includes an i7-8750H processor, 16 GB DDR4-RAM, and an Nvidia GeForce GTX 1060. The study used a fully immersive VR experience by HMD HTC^®^ Vive (HTC Co., New Taipei City, China) which tracked movements through sensors placed on the user’s body which could be easily adapted to any type of widely used RAGT device (Figure 1). HTC Vive has a resolution of 2160 × 1200, a refresh rate of up to 90 Hz, and a 110° field of view. In addition to gait rehabilitation, a RAGT helps to reduce the workload of physiotherapists [27], which is why it was essential to create a VR device easy and rapid to install and use (“plug and play” system). In addition, our VR device is quick to install, as the base stations can be directly fixed on the device (Figure 1A) or with wall mounts. Once installed, it only takes a few seconds to set up and remove from the user.

For all these reasons, we created motion in VR with the HTC Vive sensor attached to the participant’s right foot to collect the walking motion and implement synchronized movement of the avatar (Figure 1B). Whatever the orientation of the movement, the following Equation (1) brings the movement back to the path of the virtual environment (Figure 2). Longitudinal displacement is represented by the variable *x_VR_* and lateral displacement is represented through the variable *y_VR_*, which is null for use on GT. The variable *x_VR_* is used in the virtual environment to generate the displacement.
$$\{\begin{array}{c} x_{VR}=x_{GT}\times \cos\left(\theta \right)-y_{GT}\times \sin\left(\theta \right)\\ y_{VR}=x_{GT}\times \sin\left(\theta \right)+y_{GT}\times \cos\left(\theta \right) \end{array}$$
(1)$$\left(=\right)\;\left(\begin{array}{c} x_{VR}\\ y_{VR} \end{array}\right)=\left(\begin{array}{cc} \cos\left(\theta \right) & -\sin\left(\theta \right)\\ \sin\left(\theta \right) & \cos\left(\theta \right) \end{array}\right)\left(\begin{array}{c} x_{GT}\\ y_{GT} \end{array}\right),$$

The gait motion within the virtual environment is based on the participant’s walking speed and step length. The gait speed in the virtual environment has been averaged over the time it takes to complete one step. To avoid outliers, the sensor placement is compared with the original position at each step. If the sensor is no longer tracked by the base stations, the sensor’s motion velocity is too high; if the position detected is wrong, no steps to prevent cybersickness are generated in the virtual environment. With regards to the motion of the left foot, its movement is linked to the right foot with reciprocal movement. We have chosen the HTC Vive sensor, which allows simulation of gait motion using a single sensor [28] (Figure 1B) with high precision of the tracking measurements and low latency (22 ms) [29], creating a device that is easy to install and use.

The application provides a main menu for the physical therapist, giving real-time access to the virtual environment and to the patient’s clinical data (Figure 1C). The software collects the number of steps and effective walking time at each session and installs these data in a patient file that is updated at each session in the database. Step size is required to provide data on the distance walked in the virtual environment.

### 2.2. VR Environment

We used an engaging virtual environment permitting evolution in a natural space without visual distractions. The VR scenario is displayed from a first-person view (Figure 1D). We created a realistic ecological environment with a high level of graphics, using the “forest natural environment” asset in Unity (Unity Technologies^®^, San Francisco, CA, USA). To increase the RAGT time, we tried to enhance the motivation to continue with the effort by contributing a playful aspect and by assigning different targets to be seen by the patient. The participant moves in a straight line along a path through different areas (forest path, river crossing, crossroads, and city path), thereby stimulating his/her discovery of a new part of the virtual environment. At the same time, the participant listens to an audio simulation matching the environment through the HMD. When the participant looks at the ground, he can see virtual shoes displayed at the position of his/her real feet (Figure 1E).

### 2.3. Procedure

All of the healthy participants took part in two sessions of gait rehabilitation using the GT in fully immersive VR. First, each participant discovered the virtual environment in a seating position and carried out a 2-min familiarization. Then, they performed gait sessions for 20 min with 24 h separating the two sessions. They walked at 0.55 m/s and were invited to actively participate and to visually explore the environment around them. The RAGT system was activated and then stopped by the researcher. Participants wore safety harness to prevent falls. In order not to distract the participants, no verbal instructions were given once the session had started, and their behavior in the virtual environment was monitored using the computer screen (Figure 1C).

### 2.4. Measurement

Side effects, usability, and sense of presence were assessed by three questionnaires: the Simulator Sickness Questionnaire (SSQ) [30], the System Usability Scale (SUS) [31], and the Presence Questionnaire (PQ) [32]. The French version of the SSQ is a 16-item scale evaluating the severity of individual symptoms. It provides a total score and three subscales for nausea, oculomotor, and disorientation [30]. Participants completed the questionnaire before and after each session to calculate the ΔSSQ, which is the difference between the two values. To explore system usability, participants answered the SUS after the second session. It yields a score between 0 to 100 in which a higher score mean higher usability [31]. Sense of presence was assessed with the PQ after each session [32]. We used a 19-item version with a 7-point Likert scale that provided a score between 19 to 133 with a higher score indicating a higher sense of presence [32]. The number of steps and distance performed during sessions as recorded by the GT (nGT) and the fully immersive VR application (nVR) were collected.

### 2.5. Statistical Analysis

The sample size was calculated according to the previous study by Winter et al. [12] to satisfy an α level of 0.05 and a power of 0.80. Based on an increase in SSQ score above 15 to indicate the occurrence of concerning side effects [33], a minimum of 24 participants were required. SSQ, SUS, and PQ data are presented as mean with standard deviations. A Shapiro–Wilk test was performed to assess the normality distribution of the data. To determine the effect of the RAGT in fully immersive VR on the occurrence of side effects and to observe their evolution during the sessions, a Wilcoxon test was performed between pre- and post-session, as was the ΔSSQ. To investigate the evolution of the feeling of presence between the two sessions, a *t*-test was used to compare the PQ values. Statistical analysis was carried out with (XLS Stats). The significance level for alpha was set at 0.05.

## 3. Results

The age of participants was 25.2 ± 8.4, ranging from 19 to 53 years. The participants had a mean MSSQ score of 8.66 ± 6.51, and one person was not included because it was too high. All of the participants completed the two sessions of RAGT in fully immersive VR.

None of the participants reported any significant side effects requiring the interruption of a session. Twenty-nine participants (78.4%) had no significant adverse effects (SSQ < 5). Out of the eight participants with an SSQ score above 5, four had a score between 10 and 15 and three had a score above 15 (Table 1). There were no significant differences in the occurrence of side effects between the two sessions (*p* > 0.05) (Table 1). There was no significant increase in the SSQ score before and after session for session 1 (*p* = 0.203). The SSQ score increased during the second session (*p* = 0.032), but remained low, indicating no side effects. The ΔSSQ was about 2.02 (7.35) for the first session and 1.52 (4.87) for the second. The most common side effects observed were eyestrain with eight occurrences after the first session and four after the second session (Table 1).

Three people had concerning side effects with a SSQ above 15 after the first VR session (Table 1). They were female and had MSSQ scores of 25.0, 11.0, and 15.75. Only one of them had a score above 15 in the second session (26.18). Female subjects show greater susceptibility to the onset of side effects with a significantly higher MSSQ score (*p* = 0.008). That much said, there was no significant difference between men and women in terms of the onset of side effects or in increasing side effects (ΔSSQ) after the two sessions. No effect of age was found.

A significantly increased sense of presence between the two sessions was observed (*p* < 0.001) (Table 2). Out of the 37 participants, only 6 presented a decreased PQ score between the two sessions. Finally, a high level of usability was observed (86.08 ± 7.54), Table 2). Out of the 37 individuals who participated, 26 reported that the second session passed more quickly than the first.

The number of steps measured by the GT and by the VR application and the number of meters walked in the virtual environment remained constant between the two sessions with less than a 5% gap (Table 2). The VR device detected more steps than GT during both sessions (session 1: *p* = 0.005; session 2: *p* < 0.001) (Table 2).

## 4. Discussion

This study presents a new fully immersive VR application coupled with RAGT that has been developed for gait rehabilitation in non-ambulatory stroke patients. The objective of this study was to assess potential cybersickness in healthy individuals prior to investigation in non-ambulatory patients. It also aimed to observe the technological quality through the level of immersion in VR by the sense of presence and the usability of our device.

The use of RAGT combined with fully immersive VR did not lead to any cybersickness. Most of the participants (78.4%) had no or few side effects with an SSQ score below 5, which is a cut-off score for negligible symptoms [33]. None of our participants needed to stop the sessions because of side effects and only three persons had a SSQ score above 15 and corresponding to significant effects [33]. The only individuals who experienced significant side effects were women. This result seems to be corroborated by the literature [23,34]. Analysis of the different SSQ items seems to show a vision effect [35]. Some participants reported a sensation of jerky movements in moving tree leaves, which may have contributed to eye strain and headaches (Table 1). SSQ was created for a study in a flight simulator, and VR devices seems to have higher average values [23]. In comparison with immersive VR studies in neurological diseases [12,19,36], the SSQ scores show promising results suggesting good tolerance for this device. This could be explained by the direct link between the user’s movements in the real world and motion progress in the virtual environment, which can reduce sensory conflict for the user, with high realism and fluidity of movement [23]. More precisely, virtual simulation directly linked to the movements made by the user allows a high level of immersion and may decrease the risk of cybersickness [23]. The high accuracy of the HTC Vive sensor makes it an excellent tool for designing VR navigation systems in rehabilitation and should also help to prevent side effects [37]. Furthermore, we used a 3D graphic environment with low visual stimulations, which seems to be the virtual environment most likely to prevent cybersickness [23].

The onset of cybersickness is also related to the level of immersion and the user’s sense of presence in the virtual environment [18]. The few side effects observed can therefore be explained by the high sense of presence observed in this study (Table 2). We proposed a virtual environment free of distractions and as realistic as possible. In addition, we added a projection of the participants’ feet, which helped the participants to be embodied in fully immersive VR (Figure 1E). The HTC Vive HMD provides a wide field of view (110°) and can maintain high frame rates that reduce delay and lag between the physical movement and simulation. This may have facilitated the feeling of being in the virtual environment [13]. In order to increase the sense of presence, we would like to add an additional sensor on the user’s thigh to model the entire lower limb in the virtual environment and thereby promote embodiment in the avatar [38]. There was a significant increase between the two sessions regarding the feeling of presence in the virtual environment, which can be explained by a habituation effect to VR. In addition, most of the participants reported the perception of a decrease in walking time between the two sessions, which may be related to a greater sense of presence in VR. This effect may increase rehabilitation time for patients by distracting attention from the real world [12].

The use of RAGT reduces the physical workload for therapists during gait rehabilitation by providing body weight support and guidance of gait movements [27], which is why it was essential to create a fully immersive VR device that was easy to install and use. In addition, it does not require any accommodation for the patient, and its use is intuitive. This may explain the high SUS score (Table 2). At the same time, it would have been interesting to assess the usability from the therapist’s point of view. The application has a simple design and does not require special training for use. We therefore expect to find good usability by therapists in a future study with stroke patients.

### 4.1. The Future of Our Application

We created a calm and relaxing environment, which may have a positive effect on the well-being of older people and promote positive feelings, especially for non-ambulatory stroke patients who cannot go outside [39,40]. We chose a scenario without visual distractions to reduce optical flow and to minimize cybersickness [23]. That said, in order to promote active patient participation and to increase their motivation to carry out the sessions, it would be interesting to create a scenario of progression in the environment by either social interaction or by setting up intermediate objectives [9,12]. In addition, the use of cognitive tasks could lead to stronger cortical activation and thereby promote patients’ motor recovery as well as their dual-tasking abilities [11]. Fully immersive VR offers a controlled environment for the assessment of dual-task abilities during activities of daily life, especially for walking, and could reduce the dual-task cost in gait rehabilitation [41].

In the future, the coupling of portable sensors such as heart rate could provide additional information to clinicians, enabling them to more effectively tailor sessions to patient capacities and to anticipate adverse events. In addition, eye tracking could allow us to observe patients’ information acquisition and their ability to explore the environment during walking tasks. We also wish to develop an application to be used on a treadmill so that the VR application can follow the patient’s progress and improvement in walking abilities. To provide more possibilities for the creation of virtual environments, we have developed up-and-down movements that could be associated with a treadmill with a change of slope.

Although the number of steps calculated by the GT and by the VR application were consistent, there was a significant difference between the two tools (Table 2). Modifications are needed for our approach to calculate the number of steps so as to prevent overestimation of patients’ abilities. Errors in the number of steps calculated by the VR application may be due to the loss of sensor location, which could lead to overstatement in the number of detected steps. Such errors were detected and corrected in the current setup. A change in the placement of the sensor on the participant’s leg may also be considered.

### 4.2. Future Research with Stroke Patients

Following this study, which was designed to validate a new gait rehabilitation device in healthy people, we will observe its use in stroke patients, who may be more likely to develop cybersickness than healthy subjects [38]. Non-walking patients will carry out RAGT sessions with and without immersive VR and the occurrence of side effects between the two rehabilitation modalities will be observed and analyzed. In addition, we will compare the usability of the rehabilitation device with and without the fully immersive VR application. Patients’ cooperation abilities during the sessions will be considered to identify eligible patients and disorders (e.g., spatial neglect, altered cognitive abilities, and aphasia) that may limit the use of immersive VR in gait rehabilitation. A limitation of this study with healthy participants is the lack of an assessment of psychological issues prior to their participation, as certain psychological pathologies could affect participation in immersive VR (e.g., claustrophobia and anxiety). This will be evaluated to determine a standard profile of a patient who may use immersive VR during gait rehabilitation. It would also be interesting to observe the evolution of the occurrence of cybersickness over the course of sessions with stroke subjects in order to determine whether an adaptation effect is present [20]. The level of effectiveness of RAGT is directly dependent on the intensity and duration of the rehabilitation [6], which can be increased by greater motivation to perform the session [9]. Consequently, we wish to compare the level of motivation of stroke patients during a RAGT session with and without VR [9,12]. Lastly, we will compare by a controlled randomized trial the effectiveness on walking abilities of this new application in comparison to conventional electromechanical-assisted gait training and in comparison to a semi-immersive virtual reality device.

## 5. Conclusions

We presented a new VR device for gait rehabilitation of non-ambulatory patients. This study shows encouraging data regarding the technical quality of our device. The use of RAGT in combination with fully immersive VR did not lead to any cybersickness and participants experienced a strong sense of presence in the virtual environment. The low occurrence of side effects and the very good usability make it possible to design and develop future studies by exploring stroke patients’ acceptance of this device as well as the efficacy of this rehabilitation method on walking abilities compared to conventional RAGT.

## Figures and Tables

**Figure 1 medicina-58-01805-f001:**
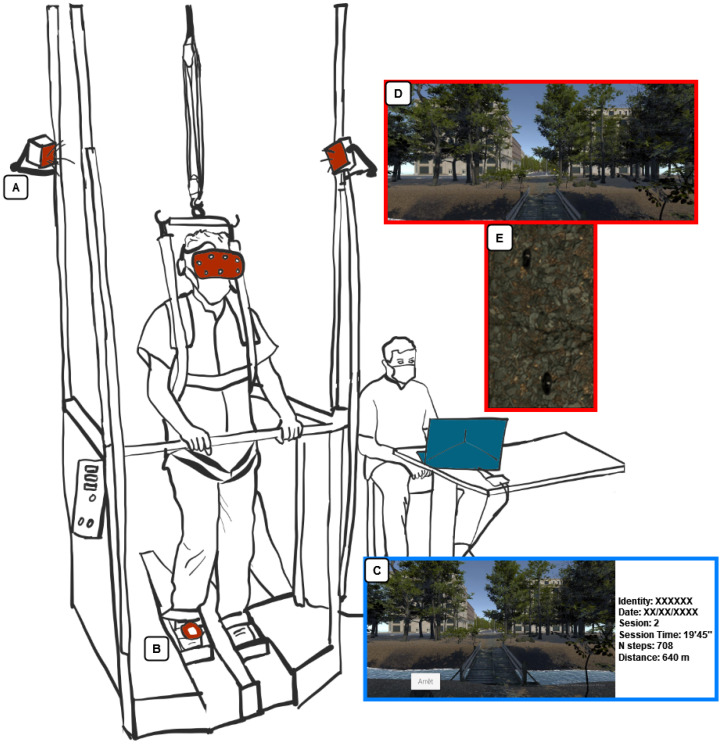
Fully immersive VR set up. (**A**) HTC Vive station base; (**B**) HTC Vive Sensor; (**C**) view of therapist; (**D**) view of participant; (**E**) participant’s feet from participant view.

**Figure 2 medicina-58-01805-f002:**
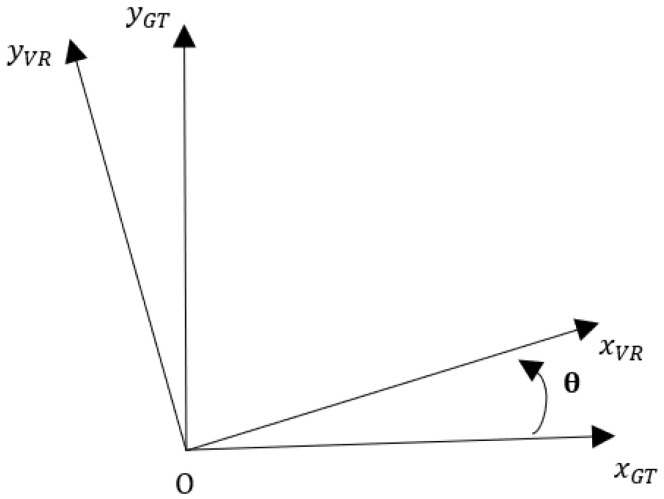
Change of reference system. *x_GT_*: physical longitudinal displacement; *x_VR_*: virtual longitudinal displacement; *y_GT_*: physical lateral displacement; *y_VR_*: lateral virtual displacement.

**Table 1 medicina-58-01805-t001:** Presentation of adverse effects by occurrence and SSQ total score.

	SSQ > 5	SSQ > 10	SSQ > 15
Individual Characteristics			
*n* (M/F)	8 (3/5)	7 (2/5)	3 (0/3)
Age (mean ± sd)	23.88 ± 3.87	24.14 ± 4.10	22 ± 1.73
MSSQ (mean ± sd)	14.63 ± 8.50	14.61 ± 9.18	17.25 ± 7.12
Occurrence of adverse effects (Session 1/Session 2)			
Eyestrain (*n*)	5 (62.5%)/3 (37.5%)	4 (57.14%)/2 (28.57%)	2 (66%)/0
Fullness of head (*n*)	3 (37.5%)/1 (12.5%)	3 (42.86)/1 (14.28%)	3 (100%)/1 (33%)
Headache (*n*)	3 (37.5%)/3 (37.5%)	3 (42.86)/2 (28.57%)	2 (66%)/0
Nausea (*n*)	3 (37.5%)/2 (25%)	3 (42.86)/2 (28.57%)	2 (66%)/1 (33%)
Vertigo (*n*)	3 (37.5%)/1 (12.5%)	3 (42.86)/1 (14.28%)	2 (66%)/0
Difficulty focusing (*n*)	2 (25%)/2 (25%)	2 (28.57%)/2 (28.57%)	2 (66%)/0
SSQ Total	Pre-session	Post-session	*p*-value
Session 1	1.31 ± 2.53	3.33 ± 6.98	0.203
Session 2	0.80 ± 2.00	2.32 ± 5.53	0.032 *

MSSQ: Motion Sickness Susceptibility Questionnaire; SSQ: Simulator Sickness Questionnaire; * *p* < 0.05.

**Table 2 medicina-58-01805-t002:** Comparison of the mean data values during RAGT sessions in fully immersive VR.

Parameters	Session 1	Session 2	*p*-Value
PQ	104.11 ± 9.37	108.73 ± 9.10	<0.001
SUS		86.08 ± 7.54	/
nGT (*n*)	704.03 ± 3.09	703.32 ± 2.33	/
nVR (*n*)	728.03 ± 49.26 *	727.24 ± 39.83 *	/
Distance in GT (meter)	675.87 ± 2.96	675.19 ± 2.24	/
Distance in VR (meter)	699.03 ± 47.32 *	698.03 ± 38.27 *	/

SSQ: Simulator Sickness Questionnaire; ΔSSQ: difference between pre- and post-session; PQ: Presence Questionnaire; SUS: System Usability Scale; nGT: number of steps measured by Gait Trainer; nVR: number of steps measured by Virtual Reality application; *: significant difference in distance and number of steps between GT and VR.

## Data Availability

The datasets used and/or analyzed during the current study are available from the corresponding author on reasonable request.

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
