# Peer review of "Immersive Virtual Reality during Robot-Assisted Gait Training: Validation of a New Device in Stroke Rehabilitation"

_medicina, 2022, doi:10.3390/medicina58121805_

Round 1

Reviewer 1 Report

This paper presents a new, fully immersive VR application for RAGT, which can be used with a head-mounted display and wearable sensors providing real-time gait motion in the virtual environment.

It seems to me that this work is an excellent contribution to the robotic rehabilitation field, it is a great work. Its improvement is suggested in terms of the following comments: 

- The resolution of the figure should be optimized.

- Please insert equation number. 

- Conclusion should be rewritten integrating the results and the future works.

- Authors must evaluate the same parameters and the effectiveness of the designed device with, at least, one stroked patient.

Concluding, the paper has potential to be appreciated by the readers and the above comment are formulated such that to enhance its impact.

Reviewer 2 Report

MAJOR COMMENTS

How did you determine the sample size?

There are many reasons for non-compliance with VR in stroke patients apart from motion sickness. Did you check if any of your participants had any psychological issue e.g. anxiety or claustrophobia? Others have difficulties in sustaining attention or have spatial neglect.

'They walked for 20 minutes in a virtual forest environment.'=>you mention this in the abstract but it is not very clear. You could maybe explain a bit more in the abstract or write this sentence clearer. Did they physically walk? Did they just see themselves walk? Or was just the environment around them moving as if they were walking?

'had no significant adverse effects (SSQ < 5)'=>It is not clear why you used those cut-offs. Could you provide a reference to explain why you used those particular ones?

By reading table 1, I see that 18/38 participants experienced at least one adverse effect. To me, this (c. 50%) seems quite high.

'We created a calm and relaxing environment, which can have a positive effect on the well-being of older people'=>but you tested young and middle-aged people, not elderly people

MINOR COMMENTS
It would be good to include dois to all references

'For research articles with several authors, a short paragraph specifying their individual contributions must be provided. The following statements should be used'=> no need to write this sentence in the contributions

'which a higher score mean higher usability'=>please correct typo

Round 2

Reviewer 2 Report

'we assume that no psychological issues were present in our tested healthy participants.'--just assuming that things happened as the investigators wished without actually asking the participants or checking their medical records is not how quality research is conducted. Thus, it is important to at least mention this in the limitations.

Regarding the SSQ and adverse effects:
You are just mentioning the severe effects in the text. It would be best to write how many participants (percentage) experienced adverse effects (any adverse effect) and then mention information regarding the severity of the adverse effects.

'We created a calm and relaxing environment, which can have a positive effect on the well-being of older people'--yes, I understand your point. Thus, you need to rephrase this sentence saying 'it may have a positive effect', because you haven't tested it yet in elderly people.

'We recruited additional participants in case of sight loss.'--you comment this but it doesn't make sense. So first you did VR and then asked about their vision and excluded them if they had vision deficits? The correct would be the opposite. I don't see in the text how you measured vision. Did you do any tests to check their vision? Which were they?

'Based on an increase of SSQ score above 15 to indicate the occurrence of side effects'--from what I understand a score above 15 shows the severity not the number of side effects, so please rephrase.

'A large majority'--please improve the use of English. The majority does mean a large portion

'Patients' cooperation abilities during the sessions will be considered to identify eligible patients and disorders (e.g., spatial neglect, altered cognitive abilities, aphasia) that may limit the use of immersive VR in gait rehabilitation'--'considered' sounds as if you won't be doing any tasks to check these disorders but you will just observe people. If you will be doing tasks then please mention which and why you chose those particular tasks.

Round 3

Reviewer 2 Report

The authors have covered my suggestions/comments. There are just a few errors in the use of English (see below).

"It will be evaluated to determine a standard profile of patient who may use immersive VR during gait rehabilitation.”--please improve the use of English, e.g. it should be A patient...

"Most of participants"--this should be THE participants